# Low-Bacterial Diet in Cancer Patients: A Systematic Review

**DOI:** 10.3390/nu15143171

**Published:** 2023-07-17

**Authors:** Sofia Matteucci, Giulia De Pasquale, Manuela Pastore, Emanuela Morenghi, Veronica Pipitone, Fanny Soekeland, Riccardo Caccialanza, Beatrice Mazzoleni, Stefano Mancin

**Affiliations:** 1IRCCS Humanitas Research Hospital, Via Manzoni 56, 20089 Rozzano, Milan, Italy; sofia.matteucci@humanitas.it (S.M.); giulia.depasquale@humanitas.it (G.D.P.); manuela.pastore@humanitas.it (M.P.); emanuela.morenghi@humanitas.it (E.M.); veronica.pipitone@gmail.com (V.P.); stefano.mancin@humanitas.it (S.M.); 2Department of Biomedical Sciences, Humanitas University, Via Rita Levi Montalcini 4, 20090 Pieve Emanuele, Milan, Italy; beatrice.mazzoleni@hunimed.eu; 3School of Health Professions, University of Applied Sciences, Schwarztorstrasse 48, 3007 Bern, Switzerland; fanny.soeke@gmail.com; 4Clinical Nutrition and Dietetics Unit, Fondazione IRCCS Policlinico San Matteo, 27100 Pavia, Italy

**Keywords:** neutropenic diet, low-bacterial diet, nutrition, cancer, hematological malignancies, safe food handling, systematic review

## Abstract

The low-bacterial diet (LBD) is a widely used dietary regimen to reduce the risk of food-borne infections in patients with neutropenic cancer, but its role is controversial due to its unclear benefits. The purpose of this study was to provide an updated analysis of the available evidence on the efficacy of the LBD to reduce the risk of infections, mortality rates, and quality of life (QoL) in neutropenic patients with cancer. A systematic literature search was conducted in the biomedical databases Cochrane Library, PubMed, CINHAL, and EMBASE. The process of the screening, selection, inclusion of articles, and assessment of risk of bias and methodological quality was conducted by two reviewers. Of the 1985 records identified, 12 were included. The LBD demonstrated heterogeneity in definition, composition, and initiation timing; moreover, the LBD did not demonstrate a reduction in infection and mortality rates compared to a free diet, showing a negative correlation with quality of life. The LBD, in addition to not bringing benefits in terms of reductions in infection and mortality rates, has been shown to worsen the quality of life due to the reduced palatability and limited variety of the food supply, negatively impacting nutritional status.

## 1. Introduction

Cancer, one of the main causes of mortality worldwide, remains a formidable challenge in the field of healthcare. Among the various types of cancer, hematologic malignancies account for approximately 6.5% [1,2,3]. These malignancies, characterized by the abnormal growth and development of blood cells, include leukemia, lymphoma, and myeloma. Patients diagnosed with hematologic malignancies often undergo aggressive treatment regimens such as chemotherapy, radiotherapy, and hematopoietic stem cell transplantation (hSCT) to combat the disease. While these therapies aim to eradicate cancer cells, they also have deleterious effects on the immune system, leading to severe bone marrow suppression [4,5,6,7].

Neutropenia refers to a condition characterized by a significant decrease in the number of neutrophils, a type of white blood cell responsible for defending the body against bacterial and fungal infections. Neutropenic cancer patients experience long-term neutropenia, which significantly increases their susceptibility to infections [4,5,6,7]. The risk is further magnified when the absolute neutrophil count (ANC) falls below 0.5 × 10^9^ neutrophils/µL. At this critical threshold, even minor infections can escalate rapidly, leading to severe morbidity and potentially life-threatening complications [4,5,8,9,10,11,12,13].

To mitigate the heightened risk of infection in neutropenic cancer patients, various preventive strategies have been employed over the years. These measures aim to minimize the exposure to infectious agents and bolster the body’s defenses. Contact precautions, including meticulous hand hygiene and the use of personal protective equipment, are implemented to reduce the transmission of pathogens. Patients may be advised to wear masks, particularly in crowded or high-risk environments, to reduce the risk of respiratory infections. Prophylactic administration of antibiotics is often recommended to prevent bacterial infections, especially during periods of severe neutropenia. Additionally, dietary restrictions have been considered an important component of infection control strategies [7,13,14].

Among the dietary interventions, the low-bacterial diet (LBD) has gained prominence as a strategy to reduce the risk of food-borne infections by limiting the introduction of potentially harmful bacteria into the gut [5,12,13,15,16,17]. The rationale behind the LBD lies in the fact that certain foods, particularly those consumed raw or minimally processed, may harbor pathogenic bacteria such as *Salmonella*, *Escherichia coli*, or *Listeria*. In a neutropenic state, even small quantities of these bacteria can lead to severe infections. Therefore, the LBD typically involves restricting the consumption of uncooked fruits and vegetables, raw fish and meat, as well as soft cheeses that may carry a higher risk of contamination. Instead, patients are advised to focus on thoroughly cooked foods, pasteurized dairy products, and commercially processed items that undergo strict quality control measures [5,12,13,15,16,17,18,19].

While the LBD has been widely implemented as a preventive measure against food-borne infections in neutropenic cancer patients, the scientific consensus regarding its benefits and practicality remains elusive, leading to ongoing debates among healthcare professionals. Nurses, as key members of the healthcare team, have a significant responsibility in promoting and educating patients about healthy behaviors and ensuring optimal nutrition. Several reputable organizations, such as the American Society for Parenteral and Enteral Nutrition (ASPEN) [20], the European Society for Clinical Nutrition and Metabolism (ESPEN) [21], the Infectious Diseases Society of America (IDSA), and the American Society of Clinical Oncology (ASCO) [22] have published guidelines on nutritional support interventions. However, when it comes to the low-bacterial diet (LBD), these guidelines reveal a lack of strong evidence supporting its efficacy. In contrast, the U.S. Food and Drug Administration (FDA) [23] has issued valuable recommendations regarding safe food handling and preparation techniques to minimize the risk of food-borne illnesses. These guidelines provide detailed information on various aspects, including purchasing safe food products, proper storage, appropriate cooking temperatures, and refrigeration practices. Such guidelines emphasize the importance of maintaining food safety throughout the entire food handling process, from purchase to consumption. However, the advice provided by healthcare providers regarding food safety can be conflicting, leading to confusion among patients and their caregivers. Consequently, the practicality and necessity of implementing a strict low-bacterial diet are being re-evaluated and actively discussed within the medical community. The ongoing debate highlights the need for further research and the development of more specific recommendations tailored to the unique needs and circumstances of neutropenic cancer patients. By gaining a deeper understanding of the benefits and drawbacks of the LBD, healthcare providers can make informed decisions and tailor dietary recommendations to best support the unique needs of neutropenic cancer patients. In the clinical field, nutrition plays a crucial role in the management of hematologic malignancies, but it poses significant challenges, particularly for patients with hematologic malignancies [12,15,19,24].

### Objective of this Systematic Review

In this systematic review, our objective is to present an updated analysis of the existing evidence regarding the method of application and the duration of administration of the low microbial diet (LBD) and its efficacy in reducing infectious risk, mortality rates, and quality of life (QoL) in neutropenic cancer patients. In terms of assessing QoL, as there are no studies utilizing specific instruments and evaluation scales, we considered the following complications which indirectly impact QoL: diarrhea, nausea, and weight loss.

## 2. Materials and Methods

### 2.1. Search Strategy

As a preliminary step, we identified relevant guidelines published by the recognized scientific society ESPEN [21]. A systematic literature search was conducted in electronic databases including PubMed, Embase, CINAHL, and Cochrane Library. The search utilized keywords such as ‘bacterial translocation’, ‘immunocompromised host’, ‘neutropenia’, and ‘low bacterial diet’, along with their variations. Boolean operators AND and OR were appropriately combined in search strings tailored to the specificities of each database. The search encompassed literature published until January 2023. The complete search algorithms can be found in the available data statement (Appendix A). The screening, selection, and inclusion process of articles adhered to the Preferred Reporting Items for Systematic Reviews and Meta-Analyses (PRISMA) statement [25]. Two review authors (S.M. and G.D.P.) independently screened all titles and abstracts identified through the electronic database searches, excluding duplications and irrelevant records using EndNote 20 software (https://endnote.com/). Conflicts were resolved through involvement of a third review author (V.P.). Full-text articles were obtained for the remaining studies and independently assessed for inclusion by two review authors (S.M. and G.D.P.) using the eligibility criteria described above. Disagreements were resolved through consensus meetings, with arbitration provided by a third review author (V.P.) who had not initially reviewed the article.

### 2.2. Registration of the Systematic Review Protocol

The protocol for this systematic review was registered in the National Institute of Health Research available at https://www.crd.york.ac.uk/prospero/ (accessed on 2 December 2022) with the PROSPERO protocol registration number CRD42022376933.

### 2.3. Research Question and Criteria

The research question was developed using the PICO methodology [26] (population, intervention, comparator, and outcome). Population: patients older than 18 years of age with hematologic malignancies; interventions: low-bacterial diet (study group—LBD); comparator: standard diet (control group—SD); primary outcomes: incidence of infection, mortality rates, and quality of life (QoL); secondary outcomes: method of application and duration of LBD. The terms low-bacterial diet, cooked diet, and neutropenic diet were used to identify the relevant studies. LBD was defined as any diet that excluded fresh raw food (such as fresh fruits and vegetables and raw meat) and other types of food such as cheese. SD was defined as any other diet used for comparison with LBD. We included both meta-analyses and observational studies published from May 2015 to January 2023, available in full text. Articles that did not meet the specified criteria were excluded.

### 2.4. Evaluation of the Risk of Bias and Methodological Quality of Studies

The risk of bias and methodological quality of the included articles were independently assessed by two reviewers (S.M. and G.D.P.) using the Joanna Briggs Institute (JBI) Critical Evaluation Tools Checklist for Systematic Reviews and Research Syntheses [27] for meta-analysis studies, Analytical Cross-Sectional Studies [28] for observational cross-sectional studies, and Analytical Case Control Studies [28] for case–control studies developed by the Joanna Briggs Institute. The complete algorithms for risk of bias and quality assessment can be found in the Appendix A.

### 2.5. Data Extraction

Two reviewers independently extracted the results from the included studies, including information such as the first author, date of publication, study design, setting, number of participants, and outcome measures. The outcome measures used in this meta-analysis were the infection rate, mortality rate, and quality of life between the study group and the control group (Appendix A).

### 2.6. Data Synthesis

The included studies were categorized into five specific objectives: application and duration of LBD, risk of infection, mortality rate, and QoL. Study designs and results were summarized for each of the five groups. The results were first presented in tables and then reported as a narrative and quantitative summary.

### 2.7. Acronyms

alloHSCT: allogeneic hematopoietic stem cell transplant; ANC: absolute neutrophil count; ASCO: American Society of Clinical Oncology; ASPEN: American Society for Parenteral and Enteral Nutrition; EBMT: European Society for Blood and Marrow Transplantation; ESPEN: European Society for Clinical Nutrition and Metabolism; FAI: food-associated infections; FDA: U.S. Food and Drug Administration; GvHD: graft-versus-host disease; hSCT: hematopoietic stem cell transplantation; IDSA: Infectious Diseases Society of America; JBI: Joanna Briggs Institute; LBD: low-bacterial diet; PICO: population, intervention, comparator and outcome; PRISMA: Preferred Reporting Items for Systematic Reviews and Meta-Analyses; QoL: quality of life; SD: standard diet.

## 3. Results

A total of 1985 articles were identified through the database searches: 46 from PubMed, 505 from Embase, 17 from CINAHL, and 1417 from the Cochrane Library. After removing 136 duplicate articles, all titles were screened, and 52 articles were retained as eligible based on reading the abstracts. Of these, 26 were deemed irrelevant, leaving 26 full-text articles for evaluation. Ultimately, 12 studies were included in this review, while 14 references were excluded. The excluded references consisted of ongoing studies, conference documents, or studies that included both pediatric and adult populations (Figure 1).

### 3.1. General Characteristics of the Included Studies

The included studies consisted of three meta-analyses [5,7,15] and nine observational studies [1,12,16,17,18,19,29,30,31] published between 2015 and 2022. The total number of participants across these studies was 3469, with 2054 in the intervention group (LBD) and 1415 in the control group (SD). The sample sizes ranged from a minimum of 18 participants to a maximum of 2086. Five observational studies employed self-reported questionnaires to analyze food restrictions related to the LBD, involving interviews with hospital centers performing hematopoietic stem cell transplantation (hSCT) [12,24,29], health professionals [19], and patients [16]. Another observational study [30] analyzed the websites of the top twenty hospitals listed in the US News Best Hospital for Cancer 2017. Additionally, three observational studies [12,18,31] evaluated the effects of implementing a free hospital diet (SD) versus the adoption of an LBD in a hospital setting to emphasize its safety in patients at high risk of developing neutropenia. Among the included studies, nine [1,12,16,18,19,29,30,31] focused on an adult population, while three [5,7,15] included both adult and pediatric patients. The latter were considered relevant to the research question, as a subgroup analysis based on age group was possible. The majority of patients included in the studies had hematologic malignancies, such as leukemia, lymphoma, and multiple myeloma, undergoing chemotherapy treatments. Additionally, four studies [1,12,24,29] provided an overview of current nutritional practices, including patients undergoing hSCT. Geographically, the studies were conducted in 27 countries, including the USA, the Netherlands, the Republic of Korea, mainland China, Italy, Spain, France, Germany, Turkey, the United Kingdom, Belgium, Poland, Switzerland, Australia, Algeria, Austria, Hungary, Croatia, the Czech Republic, Denmark, Greece, Israel, Lithuania, Norway, Russia, Sweden, and Tunisia. Five articles presented a multicenter study design [1,18,19,20,21,22,23,24,25,26] (Appendix A).

### 3.2. Application of the Low-Bacterial Diet

The literature search revealed the extensive use of various terminologies such as ‘neutropenic diet’, ‘clean diet’, ‘diet low in bacteria/microbial content’, and ‘cooked food diet’ to describe the nutritional practice, although there is no universally agreed definition [1,5,7,12,15,16,18,19,29,30,31]. Several authors define the LBD as a dietary regimen aimed at limiting the consumption of foods that can serve as a source of bacterial or yeast contamination and lead to infections associated with bacterial translocation [12,13,14,15,16].

This review [16,19,24,31] found that fewer than 50% of the 119 medical institutions surveyed had guidelines on dietary restrictions when implementing the LBD. A study [19] conducted among 110 British dietitians reported that 67.3% of the interviewed professionals provided food advice regarding the LBD to their neutropenic patients, but only 54% had protocols within their respective hospital organizations. A survey [30] conducted in the United States found that 35% of the top 20 cancer hospitals in the country provided recommendations in favor of the LBD, offering evidence to support its adoption. Brown’s survey [30] also discovered that 20% of the institutions surveyed did not recommend this clinical practice and instead provided accessible links on their websites to abstracts of published scientific articles mentioning the FDA Safe Food Handling Guidelines [23,30]. Additionally, the LBD is often implemented during the initial phase of allogeneic hematopoietic stem cell transplant (alloHSCT) as a protective measure against food-borne infections [29]. Studies conducted by Toenges et al. (2021) [29], Fang et al. (2020) [24], and Peric et al. (2018) [1] reported an incidence rate of LBD practice greater than 80% in patients undergoing hSCT. Notably, a survey [5] conducted by the European Society for Blood and Marrow Transplantation (EBMT) in 2017 revealed that the majority of centers interviewed (93%) had guidelines recommending the consumption of cooked foods. Additionally, the multicenter survey conducted by Toenges et al. (2021) [29] indicated that this practice was still considered the standard of care for patients undergoing hSCT during the neutropenic phase (86% incidence among 28 participants) and during intestinal graft-versus-host disease (GvHD) (57% incidence among 18 participants). Finally, only one study [16] included in the present systematic review examined the exposure to the LBD in the out-of-hospital setting, assessing compliance and specificity of adherence to nutritional restrictions. The analysis found that 54% of the participants followed this practice at home, while lower rates of compliance and adherence to the LBD were observed (Appendix A).

### 3.3. Duration of the Low-Bacterial Diet

Current evidence reveals conflicting criteria regarding the duration of nutritional restrictions based on ANC values, type and duration of treatment, and the patient’s immune status [1,12,19,24,29,31]. Toenges et al. (2021) [29] demonstrated that 57% of the hSCT centers interviewed (*n* = 27) recommended the LBD practice for different time periods in patients with intestinal GvHD: 1 month (4%), 3 months (32%), 6 months (21%), and 12 months (14%). Four studies [1,12,19,24,31] focused on the cutoff values of ANC used to determine the timing of the LBD practice. Carr and Halliday (2015) [19] found that 35.5% of the British dietitians interviewed prescribed the diet when the ANC value was >0.5−2 × 10^9^/L, while the survey conducted by Fang et al., (2020) [24] revealed that 27.3% of the EBMT centers interviewed discontinued the LBD practice when ANC values reached 0.5 × 10^9^/L, 45.4% at 1.0 × 10^9^/L, and 18.2% at 2 × 10^9^/L.

### 3.4. Infection Rates

The studies conducted by Ma et al. (2022) [5], Toenges (2021) [29], Jakob (2021) [18], Ball et al. (2019) [15], Heng et al. (2019) [12], and Sonbol et al. (2015) [7] aimed to investigate the clinical outcomes of the LBD and SD and their correlation with the occurrence of infectious outcomes. In this review, the total sample size considered was 2879, with 232 infectious events (15.8%) observed in patients from the intervention group (*n* = 1464) and 188 (13.2%) in the control group (*n* = 1415). Two articles [5,15] reported comparable rates between the two study groups regarding the risk of developing infections, including bacteremia, fungaemia, and pneumonia. These rates were determined through clinical examinations in only two studies [12,18] and diagnostic tests, such as blood cultures and stool samples. Specifically, Jakob et al. (2021) [18] documented a total of zero positive blood cultures (0%) and four positive stool samples (0.4%) in the intervention group (*n* = 1043), while the control group (*n* = 1043) had four positive blood cultures (0.4%) and three positive stool samples (0.3%). Heng et al. (2019) [12] reported 40 bloodstream infections (50.6%) in the intervention group (*n* = 79) and 25 (33.3%) in the control group (*n* = 75). The main pathogenic organisms identified were *Campylobacter coli*, *Bacillus cereus*, *Salmonella typhimurium*, hydrophilic *Aeromonas*, and *Staphylococcus aureus*, affecting various areas such as blood, respiratory, intestinal, and urinary sites [12,18]. Lastly, a study [29] reported that food-associated infections (FAI) occurred more frequently in the home setting (64%) compared to the hospital setting (18%) (Table 1).

### 3.5. Mortality Rates

None of the included studies specifically evaluated factors associated with infection-related mortality rates. However, two meta-analyses [5,7] reported comparable rates between the two study groups. Among the total sample size considered (*n* = 2392) in this review, a mortality rate of 8.6% was observed in patients in the intervention group (*n* = 1199), while the control group had a mortality rate of 9.2% (*n* = 1193). It is important to note that the study conducted by Jakob et al. (2021) [18] observed a higher number of deaths in the control group compared to the intervention group. However, these deaths were not associated with food-borne infections but were caused by other factors (Table 2).

### 3.6. Quality of Life

Quality of life was assessed in a single study [18], which examined the impact of the two nutritional practices (LBD and SD) on the incidence of diarrhea (defined as three or more loose or watery stools per day), nausea, and weight loss exceeding 1 kg. The study found that the intervention group experienced a higher prevalence of complications compared to the control group: diarrhea (RR = 0.80, 95% CI 0.73–0.89; *p* < 0.001), nausea (RR = 0.85, 95% CI 0.80–0.90; *p* < 0.001), and weight loss (RR = 0.93, 95% CI 0.86–1.00; *p* = 0.050) (Table 3).

## 4. Discussion

After conducting a comprehensive evaluation of the available literature, it became increasingly evident that the low-bacterial diet imposes significant limitations on various fresh food products, including fruits, vegetables, meat, and fish. These dietary restrictions, although initially designed to protect immunocompromised patients from potential infections, pose considerable risks without clear clinical evidence supporting their introduction. Comparative studies conducted between patients following the LBD and those adhering to a free diet indicate similar rates of infection and mortality.

In addition, it is worth noting a study [2] that was not included in the review due to methodological concerns. This study, despite its limitations, demonstrated an increased risk of infections and diarrhea in patients following LBD restrictions. Conversely, investigations focusing on the implementation of a free diet (SD) have consistently demonstrated its safety for patients and its potential role as a protective factor against common issues such as diarrhea, nausea, and weight loss. These studies have examined the effects of a well-balanced diet that emphasizes food safety practices, including proper handling, storage, and cooking techniques to minimize the risk of food-borne infections. The implementation of an SD not only addresses the concerns associated with the LBD but also allows patients to enjoy a broader range of food choices, ensuring a more satisfying and nutritionally adequate diet. The limitations imposed by the LBD extend beyond the restricted food choices and their impact on nutritional intake. They also hamper the overall quality of life of immunocompromised patients. It is important to consider that these patients often face additional challenges related to their underlying condition, including physical and psychological stress, reduced appetite, and changes in taste preferences. The LBD, with its strict preparation rules and food restrictions, can further exacerbate these challenges, potentially leading to inadequate nutrient intake and compromised well-being. Moreover, the nutritional status of immunocompromised patients is a critical factor that needs to be taken into account when considering dietary interventions. Patients undergoing cytotoxicity treatments often experience malnutrition [32] due to factors like malabsorption, food aversion, and the metabolic demands of the disease itself. For these patients, a diet that supports optimal nutrient intake is crucial for maintaining their strength, promoting recovery, and minimizing treatment-related complications. The strict cooking methods associated with the LBD can have unintended consequences on nutrient availability and bioavailability. Cooking techniques such as boiling, baking, and soaking can result in nutrient loss, particularly heat-sensitive vitamins and minerals. Micronutrients play a vital role in supporting immune function and overall health, making it essential to ensure their adequate intake in immunocompromised patients. Furthermore, the dietary restrictions imposed by the LBD limit the consumption of fibers, vitamins, and minerals that are already deficient in this specific patient population. Fiber, for instance, is known to maintain intestinal integrity, regulate bowel movements, and support a healthy gut microbiome. Insufficient fiber intake can lead to gastrointestinal disturbances and compromise the overall health of the patients. Similarly, micronutrient deficiencies can impair immune restoration and increase the risk of complications such as GvHD in patients undergoing transplantation.

It is crucial to consider cultural factors when evaluating the implications of the LBD. In certain cultures, the dietary restrictions imposed by the LBD may eliminate key components of daily food intake. For example, fresh fruits such as figs and dates are integral parts of some diets, providing important nutrients and culinary enjoyment. Raw meat and vegetables are also commonly consumed in certain cuisines, and their exclusion may significantly impact the cultural and sensory aspects of meals. Therefore, it becomes imperative to identify an optimal food program that ensures the best possible quality of life for this specific patient population [33] while respecting their cultural preferences and dietary traditions.

In light of these concerns and limitations, it is important to note that the current guidelines from the EBMT do not recommend the LBD. Similarly, US health organizations have already issued recommendations to replace this practice with safe food handling guidelines [23], aiming to reduce the potential risk of food-borne infections. However, it is worth highlighting that significant discrepancies and non-compliance with national and international guidelines exist, as previously emphasized by published studies. This disparity reflects a common problem faced by numerous institutions worldwide, highlighting the need for improved awareness and adherence to evidence-based guidelines.

The general characteristics of the included studies revealed a diverse range of research designs and populations, contributing to the robustness and applicability of the findings.

The main limitations of this review reside in the different methodological quality between the various included studies, and in particular the meta-analyses of randomized controlled trials have provided the best available evidence for the analysis of the presented clinical results. There are further limitations in the methodology and in the study designs, mainly referring to the primary studies considered. One limitation is the use of non-validated questionnaires with reduced response rates to the proposed surveys. Another is the difficult assessment of relevant outcomes, such as the impact of the SD and LBD on patients’ well-being, physical health, and psyche. Although the included studies cited the importance of nutrition in quality of life, only a small number of studies analyzed this aspect. Furthermore, the lack of a standardized definition of the LBD complicates the interpretation of the results, despite the general agreement regarding the exclusion of unpasteurized dairy products, fruits, vegetables, and raw or undercooked meat and fish.

However, it is important to acknowledge that the overall quality of the included studies varied, with differences in methodology, sample sizes, and outcome measures.

The inclusion of meta-analyses of randomized controlled trials provided the best available evidence, while the observational studies provide insights into real-world practices and outcomes.

The total number of participants across these studies was 3469, with varying sample sizes, reflecting the heterogeneity of the patient populations included in the research. The majority of studies focused on the adult population, particularly patients with hematologic malignancies undergoing chemotherapy treatments, which is a high-risk group for developing neutropenia. This focus on specific patient populations enhances the relevance of the findings to clinical practice. Additionally, the geographical distribution of the studies in 27 countries demonstrates the global interest in this dietary regimen. It is worth noting that five articles presented a multicenter study design, which further strengthens the generalizability of the findings and allows for a broader understanding of the LBD’s application in different healthcare settings.

## 5. Conclusions

The present systematic literature review provides an overview of the application and outcomes of the low-bacterial diet in patients with hematological malignancies at risk of developing neutropenia. The review findings emphasize the different characteristics and practices related to the LBD, reflecting the global interest in this dietary regimen. The studies included in this review shed light on the limitations of the LBD, which is characterized by reduced palatability and a limited variety of food supply. These factors contribute to malnutrition among patients with severe immune impairment, ultimately impacting their quality of life. Moreover, despite the strict dietary restrictions imposed by the LBD, it does not guarantee significant reductions in food-borne infections or related mortality rates. Recognizing the pivotal role of nutrition in the successful treatment of cancer and its positive effects on patient survival and prognosis, various guidelines [20,21,22,23] emphasize the significance of addressing the nutritional needs of these individuals. Therefore, it becomes crucial to provide specific education and guidance on adopting a safe hospital/home diet while ensuring the safe handling and preparation of high-risk foods based on FDA-approved food safety guidelines [23]. In this regard, it should be noted that in a study [29] included in this review, the infection rates associated with food were higher in the home and outpatient settings than in the hospital setting. These data could be attributed to the fact that in a healthcare environment, adherence to FDA guidelines would reduce food contamination, ensuring greater dietary safety. In addition to the nutritional benefits, a free diet offers psychological and emotional advantages for patients. It allows them to have more autonomy and control over their food choices, which can positively impact their overall satisfaction and well-being during the treatment process. By having a wider range of food options available, patients can find pleasure in eating, which may help improve their appetite and nutritional intake. Furthermore, the provision of specific education empowers patients and their caregivers with the knowledge and skills to handle food safely. Patients are educated on proper hygiene practices, such as thorough handwashing and appropriate cooking temperatures, to reduce the likelihood of complications arising from food-borne illnesses. This approach promotes a holistic approach to patient care by considering not only the nutritional aspects but also the psychological and emotional well-being of individuals undergoing treatment. In conclusion, this systematic literature review underscores the positive effects of a free diet over an LBD for patients with hematological malignancies. It highlights the need for quality- and safety-based nutritional assistance, as well as specific education regarding the adoption of safe food handling practices [23]. By doing so, healthcare professionals can minimize non-essential dietary restrictions and enhance the nutritional support provided to a vulnerable population at high risk of malnutrition, ultimately improving their treatment outcomes, quality of life, and overall satisfaction during their cancer journey.

## Figures and Tables

**Figure 1 nutrients-15-03171-f001:**
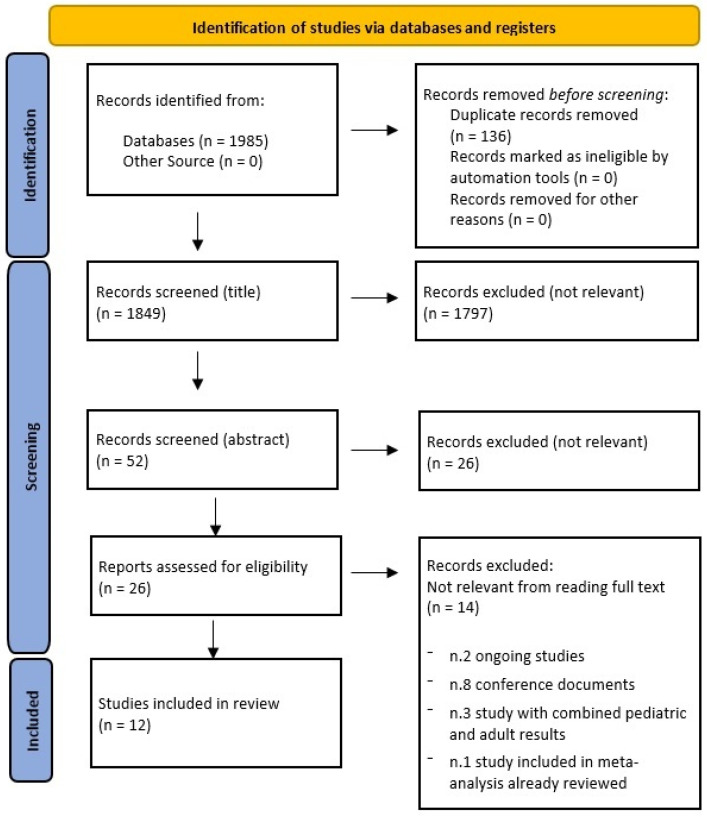
PRISMA flow chart of the process of article inclusion.

**Table 1 nutrients-15-03171-t001:** Infectious rates.

Study	Design	Sample (*n*)	Study Group (ND)	Control Group (SD)
Ma et al. [5]	Meta-analysis	LBD: 113 SD: 106	67.3%	62.3%
Toenges et al. [29]	Cross-sectional	LBD: 28	82.1%	//
Jakob et al. [18]	Case–control	LBD: 1043 SD: 1043	0.2%	0.35%
Ball et al. [15]	Meta-analysis	LBD: 113 SD: 106	67.3%	62.3%
Heng et al. [12]	Cross-sectional	LBD: 79 SD: 75	58.2%	42.7%
Sonbol et al. [7]	Meta-analysis	LBD: 88 SD: 85	8%	20%

Abbreviations: LBD = low-bacterial diet; SD = standard diet.

**Table 2 nutrients-15-03171-t002:** Mortality Rate.

Study	Design	Sample (*n*)	Study Group (ND)	Control Group (SD)
Ma et al. [5]	Meta-analysis	LBD: 113 SD: 106	43.6%	38.7%
Jakob et al. [18]	Case–control	LBD: 1043 SD: 1043	3.3%	5%
Sonbol et al. [7]	Meta-analysis	LBD: 88 SD: 85	43.6%%	38.7%

Abbreviations: LBD = low-bacterial diet; SD = standard diet.

**Table 3 nutrients-15-03171-t003:** Quality of life.

Study	Design	Sample (*n*)	Study Group (ND)	Control Group (SD)
Jakob et al. [18]	Case–control	LBD: 1043 SD: 1043	Diarrhea: 51.1% Nausea: 74.6% Weight loss: 64.8%	Diarrhea: 40.2% Nausea: 63.3% Weight loss: 60.2%

Abbreviations: LBD = low-bacterial diet; SD = standard diet.

## Data Availability

Not applicable.

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
