# Peer review of "Low-Bacterial Diet in Cancer Patients: A Systematic Review"

_nutrients, 2023, doi:10.3390/nu15143171_

Round 1

Reviewer 1 Report

Dear authors,

 I would like to thank you for the opportunity to review this manuscript.

This study, which investigated how the diet poor in bacteria reduces the risk of

food infections in neutropenic cancer patients, is very interesting.

Here are my comments:

Study design, data collection and analysis were performed appropriately.

The supplementary materials were very helpful.

The description of the method is extensive and detailed.

There should be a more detailed description of the LBD diet that was considered in the study.

Chapter 3.3 deals with the duration of the low carbohydrate diet or the duration of the LBD?

It is interesting to note if the LBD diet patients received the necessary nutrient intake artificially (supplements, vitamins, etc.)?

The conclusions chapter will highlight more clearly whether the purpose of the study has been achieved.

Please insert a chapter with abbreviations

I wish you good luck!

Author Response

Dear Prof. Dr. Lluís Serra-Majem and Prof. Dr. Maria Luz Fernandez,

Editors-in-Chief of Nutrients

Thank you for giving us the opportunity to submit a revised draft of our manuscript titled “Low bacterial diet in cancer patients: a systematic review”. We appreciate the time and effort that you and the Reviewers have dedicated to providing your valuable feedback on our manuscript. We are grateful to the Reviewers for their insightful comments on our paper.

We have successfully incorporated the suggested changes provided by the reviewers and have expanded the word count in the text to meet the criteria outlined by your journal. Here is a point-by-point response to the Reviewer's comments and concerns.

Reviewer 2 Report

The manuscript by Matteucci et al. provided valuable insights for the controversial role of low bacterial diet (LBD) in patients specifically with hematologic cancers. The authors conducted a systematic literature review and analysis based on a comprehensive range of sources of studies. The study was presented with a clear flow of logic for its background/purpose and thorough description of the definition and implementation of LBD (along with its limitations). The authors also discussed the bias and drawbacks of the current study, providing reasonable limitations of the conclusion made from the analysis with the given material. Overall, the manuscript serves the readers well by providing meaningful suggestions that the LBD might not provide clear benefit for immunocompromised cancer patients, and certain LBD might impose negative effects on the patients’ quality of life. Though with its discussed limitations, these information is rather useful to be taken into consideration for clinical practice for patients in the studied categories or be referred in future studies. However, there are a few minor points that requires improvements or clarification that will help to improve the quality of this article:

1.     The resolution of Figure 1 needs to be improved.

2.     The authors mentioned in Result “3.3 Duration of the low carbohydrate diet” without a clear discussion of why introducing the “low carbohydrate diet” is relevant to the current topic. A clear explanation of this is needed to justify this section in the result.

3.     Although the main conclusion based on the analytical result of this study did not indicate a significant benefit of LBD, the authors mentioned that in one study (in Result “3.4 Infection rates”), food-associated infections occurred more frequently in the home setting than in a hospital setting. Therefore, it will add valuable weight to the discussion of this study if the authors could include a further elaboration on this point, such as, if the results in other studies reviewed by this study were affected by the setting of the implementation of the LBD. As the authors already mentioned in “Result 3.2 Application of the low-bacterial diet”, only one study included in the present review showed exposure to the LBD in the out-of-hospital setting. The authors could easily make links of these two statement and also provide necessary details to make this point clear for the readers.

4.     It might still be helpful to provide more information in the cause of death for the Jakob et al. study in “Result 3.5 Mortality rates”, so that the readers could decide if the “other causes” demonstrate any confounding or secondary causes to the association with food-borne infections.

5.     The phrase “LBD diet” occurred several times in the manuscript. The acronym “LBD” already contain the word “diet” with the “D”. Therefore, the authors do not need to use the word “diet” after “LBD”.

The quality of English language is very good with the need of minor editing. 

Author Response

(The authors gave the same response as above.)
